# Biochar Improves Yield by Reducing Saline–Alkaline Stress, Enhancing Filling Rate of Rice in Soda Saline–Alkaline Paddy Fields

**DOI:** 10.3390/plants13162237

**Published:** 2024-08-12

**Authors:** Weikang Che, Xuebin Li, Junlong Piao, Yue Zhang, Shihao Miao, Hongyue Wang, Liming Xie, Feng Jin

**Affiliations:** Agronomy College, Jilin Agricultural University, Changchun 130118, China; 13154369230@163.com (W.C.); lixuebin91666@163.com (X.L.); ppp2147353805@163.com (J.P.); 15044554600@163.com (Y.Z.); miaoshihao2024@163.com (S.M.); 15500279917@163.com (H.W.); 15043632826@163.com (L.X.)

**Keywords:** saline–alkaline stress, biochar, grain filling, physiological indicator, enzyme activity, rice

## Abstract

Soda saline–alkaline stress significantly impedes the rice grain filling process and ultimately impacts rice yield. Biochar has been shown to mitigate the negative impacts of saline–alkaline stress on plants. However, the exact mechanism by which biochar influences the rice grain-filling rate in soda saline–alkaline soil is still not fully understood. A two-year field experiment was conducted with two nitrogen fertilizer levels (0 and 225 kg ha^−1^) and five biochar application rates [0% (B0), 0.5% (B1), 1.5% (B2), 3.0% (B3), and 4.5% (B4) biochar, *w*/*w*]. The results demonstrated that biochar had a significant impact on reducing the Na^+^ concentration and Na^+^/K^+^ ratio in rice grown in soda saline–alkaline lands, while also improving its stress physiological conditions. B1, B2, B3, and B4 showed a notable increase in the average grain-filling rate by 5.76%, 6.59%, 9.80%, and 10.79%, respectively, compared to B0; the time to reach the maximum grain-filling rate and the maximum grain weight saw increases ranging from 6.02% to 12.47% and from 7.85% to 14.68%, respectively. Meanwhile, biochar, particularly when used in conjunction with nitrogen fertilizer, notably enhanced the activities of sucrose synthase (SuSase), ADPG pyrophosphorylase (AGPase), starch synthase (StSase), and starch branching enzyme (SBE) of rice grains in soda saline–alkaline lands. Furthermore, rice yield increased by 11.95–42.74% in the B1, B2, B3, and B4 treatments compared to the B0 treatment. These findings showed that biochar improves yield by regulating ionic balance, physiological indicators, starch synthesis key enzyme activities, and the grain-filling rate in soda saline–alkaline paddy fields.

## 1. Introduction

Saline–alkaline stress is among the abiotic stresses that severely inhibit plant growth and development in the world. The Songnen Plain in Northeast China is located between 42 30′–51 20′ N and 121 40′–128 30′ E, and is a major s saline–alkaline area with an area of approximately 3.42 million hectares [1]. The main components of these problem soils are NaCO_3_ and NaHCO_3_, which are poorly structured and nutrient-poor. It contains excessive salinity, which reduces water potential, making it difficult for crops to absorb water from the soil [2,3]. In addition, excessive accumulation of Na^+^ in plant cells not only causes physiological metabolic disorders in the cells but also antagonizes the uptake of nutrients and potassium ions, thus disrupting the sodium–potassium ion balance in the cytoplasm and causing ionic toxicity, which leads to nutritional imbalance [4,5,6]. Under soda saline–alkaline stress, a large amount of reactive oxygen species (ROS) are produced in cells, destroying the cell structure, causing metabolic disorders in cells, and triggering oxidative damage [7]. Meanwhile, high pH, along with osmotic stress and ionic toxicity, can significantly harm cell membrane structure, decrease crop root vigor and photosynthetic function, ultimately leading to reduced yield [8]. Research indicates that the water needed for rice cultivation can help leach salts from the soil, thereby potentially alleviating the negative impacts of soda saline–alkaline stress [9,10]. However, with changing environmental conditions, soda saline–alkaline paddy soils are encountering challenges like water scarcity and escalating soil salinization. These issues significantly impact the growth, development, and yield of rice seeds [11,12,13]. Hence, enhancing the structure of soda saline–alkaline soils and mitigating barriers in such areas are crucial for advancing sustainable development.

The crop filling stage is a key developmental stage for yield production and is complexly modulated by environmental factors and inherent genetic machinery [14,15]. The grain filling process is a dynamic change in the accumulation pool of photosynthetic products, which are transported to the grain as sucrose and converted to starch for storage in the grain [16]. It has been shown that saline–alkaline lands are structurally unstable and nutrient-poor, and their low soil contents of nitrogen, phosphorus, potassium, and organic matter significantly reduce soil nutrient uptake by the crop root system, leading to crop nutrient deprivation and reduced photosynthetic capacity [17,18]. Previous studies have found that ionic toxicity caused by higher Na^+^ concentrations in saline–alkaline soil leads to nutrient imbalances in rice leaves that limit photosynthesis and inhibit reservoir size [19], and that nutrient imbalances also restrict crop absorption of necessary nutrients and water, destroying chloroplast structure and reducing chlorophyll content, leading to a decrease in photosynthetic efficiency [20]. Meanwhile, salt stress significantly increases the accumulation of ROS and inhibits the transport and utilization of photosynthetic products, resulting in insufficient photosynthetic products in rice [7]. Lower photosynthetic capacity limits panicle differentiation, prolonging the time of grain filling, and significantly decreasing the filling rate and grain weight of the grains [21,22,23]. Liu et al. [24] found that high concentrations of ABA in seeds under saline–alkaline stress inhibited the excessive accumulation of ROS, reduced membrane damage, and consequently limited starch synthesis under pot conditions with the addition of NaCl solution. In addition, saline–alkaline stress leads to a significant reduction in crop yield by inhibiting rice panicle formation and fertilization processes [25,26]. Cui et al. [13] added sea salt to the soil in a pot experiment and found that an increase in the proportion of long chains in branched-chain starch in the seeds under heavy salt stress led to a decrease in the activities of isoenzymes of starch synthase (*SSI*, *SSII*, and *SSIII*), inhibiting the synthesis of starch. Peng et al. [27] showed that salinity stress reduced the activities of SuSase, AGPase, and StSase in seeds by adding 0.6% sea salt solution after rice flowering, mainly because salinity stress significantly reduced the coding genes *OsSUS2*, *OsSUS3*, *OsSUS4*, *OsSSI*, *OsSSIIIa*, and *OsAGPS2* in rice grains expression, negatively affecting the sugar dynamics of the source and developing reservoir tissues, which in turn inhibited rice filling efficiency. Thus, the abatement of saline–alkaline stress and the synergistic enhancement of the irrigation efficiency of saline rice are important technological approaches to improve the yield of rice in saline–alkaline paddy fields.

Biochar is a solid product derived from organic material, subjected to the process of non-oxidative thermal decomposition [28]. Biochar is a carbon-rich material that helps sequester atmospheric carbon in the soil and improves the soil’s physicochemical properties and microbial activity [17,24]. In addition, biochar holds promising potential as a soil amendment for the revitalization of salt-affected soils [29]. Numerous studies have demonstrated that biochar’s superior structural and physicochemical qualities can reduce leaf Na^+^ accumulation, the Na^+^/K^+^ ratio, and the effectiveness of K^+^ in improving leaf water status, thereby mitigating ion toxicity and osmotic stress in crops [11,30]. Using a pot experiment that combined biochar with phosphate fertilizer, El-Desouki et al. [18] discovered that biochar increases the growth of oilseed rape in acidic soils by boosting the electron transfer rate and photosynthetic rate and lowering the malondialdehyde content. In a pot experiment, Gong et al. [31] discovered that, by boosting the expression of genes including *SSI*, *SSII*, *SBEI*, and *SBEIb*, rice husk biochar was able to dramatically increase the activity of StSase, SBE, and DBE during the rice filling stage. In the Songnen Plain’s soda saline–alkaline fields, biochar treatment improved the structure and function of maize roots by strengthening grain-filling rate, boosting nutrient and water uptake, and improving saline soil structure and nutrient effectiveness. These actions ultimately increased maize dry matter accumulation and grain yield [32]. Cui et al. [33] applied straw biochar to coastal saline–alkaline soils and found that the biochar was able to enhance wheat and maize yields by improving the physicochemical properties and microstructure of coastal saline soils, increasing soil water-holding capacity, total organic carbon content, effective potassium, and cation exchange capacity. Previous investigations conducted by our research team have demonstrated that biochar significantly affected the relative abundance of bacterial communities and altered the bacterial community structure by significantly decreasing the BD, ESP, ECe, and SARe, and increasing the soil CEC, TP, Ks, and SOM in soda saline–alkaline land [34,35]. At the same time, biochar optimized the soda saline–alkaline paddy soil rice root structure and improved root uptake capacity [23]. Piao et al. [19] showed that peanut hull biochar promoted leaf photosynthetic rate and consequently net assimilation rate of rice by increasing chlorophyll concentration, leaf N concentration, LAI, Gs, and Tr in a soda saline–alkaline paddy field. Currently, most studies have focused on the improvement of salt-stressed soil structure, soil fertility enhancement, and saline–alkaline barrier abatement by biochar, and most of them are in pots and controlled experiments, but there is no systematic report on the study of the mechanism of how biochar regulates the ion balance of rice organs, the physiological indicators, the activity of key enzymes of grain starch synthesis, or the rate of grain filling to improve the yield under the field conditions of soda saline–alkaline lands.

Our study hypothesized that biochar can increase the grain-filling rate and abate the adverse effects of soda saline–alkaline barriers on rice growth by regulating the ionic balance, physiological indicators, and activity of key enzymes of starch synthesis in soda saline–alkaline paddy fields. Therefore, this experiment was conducted through a field partitioning trial with different nitrogen fertilizer gradients and biochar application rates in soda saline–alkaline paddy lands. The effects of biochar on ionic balance, physiological indices, grain-filling characteristics, and yield of rice in soda saline–alkaline soils were investigated. In order to reveal the potential regulatory mechanisms of how biochar can enhance rice yield by regulating rice grain-filling parameters. The results of the study have important theoretical and practical significance for promoting the sustainable development of rice farming in soda saline–alkaline lands to ensure ecological environment and food security.

## 2. Materials and Methods

### 2.1. Experimental Design

The experiment described in this study was conducted at the Saline–Alkaline Experimental Site of Jilin Agricultural University in Baicheng City, Jilin Province, Northeastern China, during the period of 2022–2023. The geographical coordinates of the experimental site are 45°35′ N, 123°50′ E. The region experiences an average annual temperature of 5.2 °C, average annual precipitation of 399.9 mm, average annual evapotranspiration of 1840 mm, and an average active cumulus of 2996.2 °C (2022–2023). According to the United States Department of Agriculture (USDA) texture classification system, the soil in the test area was soda saline–alkaline [36]. The test field had been plowed for 4 years before the test. The physicochemical properties of the soil at the test site are shown in Table 1. The experiment was a two-factor partitioned layout, with nitrogen fertilizer as the major area and biochar as the sub-area, with a total of 30 plots in three replications, with a plot area of 15 m^2^ (3 × 5 m), separated by row spacing (60 cm wide), and with an individual drainage and irrigation system for each plot. The experiment was set up with two nitrogen fertilizer treatments and five biochar application treatments. The nitrogen fertilizer gradient was no nitrogen application (N0) and regular nitrogen application (N225). The biochar gradients were B0 (0.0% *w*/*w*), B1 (0.5% *w*/*w*), B2 (1.5% *w*/*w*), B3 (3.0% *w*/*w*), and B4 (4.5% *w*/*w*), which corresponded to field application rates of 0 t ha^−1^, 6.75 t ha^−1^, 20.25 t ha^−1^, 40.50 t ha^−1^, and 60.75 t ha^−1^, respectively.

The test variety was Changbai 9, which was sown in the greenhouse on 12 April 2022 and 15 April 2023 and transplanted to the paddy field on May 21st and May 25th. The transplanting density was 30 cm × 16.5 cm, with 3 plants per hole. Harvest was conducted on 30 September 2022, and 1 October 2023. Measures of 225 kg N ha^−1^, 70 kg P ha^−1^, and 75 kg K ha^−1^ were applied as fertilizer for N225, while 0 kg N ha^−1^, 70 kg P ha^−1^, and 75 kg K ha^−1^ were applied as fertilizers for N0. Other field practices are the same as local cultivation methods.

### 2.2. The Properties of Biochar

The biochar was obtained from peanut shells through a pyrolysis process conducted without oxygen at 450 °C for 4 h. It was procured from Lusen Carbon Powder Technology Co. Ltd. in Qiqihaer, China. The physical and chemical properties of the used biochar are shown in Table 2. In April 2022 and April 2023, biochar was applied to the soil surface and subsequently uniformly blended into the 0–20 cm soil horizon.

### 2.3. Sampling and Measurements

#### 2.3.1. Measurement of Leaf Concentration of Na^+^, K^+^, and Na^+^/K^+^ Ratio

At maturity, all the leaves of six rice plants from each treatment were gathered, dried in an oven at 70 °C until constant weight, and then milled into powder. The digested leaf samples were treated with 1% perchloric acid nitrate, according to the method described by Bastías et al. [37]. The concentrations of Na^+^ and K^+^ were determined using the flame photometric method (M410, Sherwood Scientific Ltd., Cambridge, UK). A measure of 1 mg mL^−1^ Na^+^ standard solution (0 mL, 1 mL, 2 mL, 3 mL, 4 mL, and 5 mL) and K^+^ standard solution (0 mL, 0.2 mL, 0.4 mL, 0.6 mL, 0.8 mL, and 1.0 mL) were added in six test tubes. The data were read under a flame photometer and plotted as a standard curve from which the Na^+^, K^+^ concentrations, and Na^+^/K^+^ ratios were calculated.

#### 2.3.2. Sugar-Soluble Protein and Proline

Three representative raphe leaves of each rice plant were selected at the heading stage and filling stage of rice and were frozen in liquid nitrogen and stored in a refrigerator at −80 °C for the measurement of physiological indexes. Estimation of soluble sugars was conducted according to the phenol–sulfuric acid method [38]: Take 6 numbered test tubes and add 0–1.0 mL of glucose solution (0 mL, 0.2 mL, 0.4 mL, 0.6 mL, 0.8 mL, and 1.0 mL) and distilled water (1.0 mL, 0.8 mL, 0.6 mL, 0.4 mL, 0.2 mL, and 0 mL) and then phenol solution in turn. Add 5 mL of concentrated sulfuric acid in 5–20 s and shake well. The standard curve was plotted with glucose content as horizontal coordinate and absorbance as vertical coordinate. Absorbance values were determined at 620 nm using glucose as a standard. Soluble sugar content was determined as mg g^−1^ F W, using a calibration curve. The determination of soluble protein content in plant samples was conducted utilizing the Khomas Brilliant Blue G-250 method, as described by Bradford [39]. Firstly, 6 test tubes were taken and 0–1.0 mL of standard protein (100 μg mL^−1^) and distilled water were added sequentially; then, 5 mL of Khomas Brilliant Blue G-250 was added and shaken well and then left for 2 min. The standard curve was plotted with the protein content as the horizontal coordinate and the absorbance, measured at 595 nm, as the vertical coordinate. Then, 0.5 g of the sample was homogenized in 10 mL of phosphate buffer pH 7.8 and centrifuged at 10,000 rpm for 20 min. After centrifugation, 0.1 mL of protein extract was mixed with 0.9 mL of tri-HCl buffer solution and 5 mL of G-250 Khomas Reagent and allowed to stand for 2 min at room temperature. The absorbance of the supernatant was measured at 595 nm on a spectrophotometer. For proline, in 6 test tubes, 2 mL of standard concentration of proline solution (concentration of 1–6 μg mL^−1^, respectively) was added, 2 mL of glacial acetic acid and 2 mL of acidic ninhydrin solution were added sequentially, and the mixture was heated in a boiling water bath for 30 min; after cooling, 4 mL of toluene was added, and this was left to stand after oscillating for 30 s. The standard curve was drawn after colorimetry at 520 nm with toluene solution as blank control. A measure of 0.5 g of the crop sample with 5 mL of 3% sulfosalicylic acid in a boiling water bath for 10 min. Following the extraction, 2 mL of the heated extract was allowed to cool and was subsequently mixed with 2 mL of glacial acetic acid and 2 mL of ninhydrin. The resultant mixture was subjected to boiling for a further 30 min then cooled, and it was extracted with toluene. The absorbance of the toluene fraction from the upper layer was measured at 520 nm, employing the method described by Bates et al. [40]. Proline content was determined by the calibration curve and expressed as μg g^−1^ FW.

#### 2.3.3. Malondialdehyde (MDA), Superoxide Anions (O_2_^−^), and Hydrogen Peroxide (H_2_O_2_)

Stored fresh leaves were removed and MDA content (nmol g^−1^ FW) was determined depending on the method of Stewart and Bewley by the supernatant, obtained after centrifugation of the homogenate was mixed with 2 mL of 0.67% thiobarbituric acid (TBA) and subjected to a heating process at 100 °C for 30 min, followed by cooling with ice water; then, the centrifuging procedure was conducted at 4000 rpm for 10 min. The resulting supernatant was then evaluated for absorbance at 450, 532, and 600 nm, respectively. The concentration of MDA was measured according to the following formula:MDA (nmol g^−1^ FW) = [6.45 × (A532 − A600) − (0.56 × A450)] × V^−1^ W
where V = volume (mL); W = weight (g) [41].

The O_2_^−^ content (μg g^−1^ FW) was measured by the method of Wang and Jiao [42]. NO_2_^−^ (5 μg mL^−1^) standard solution (0 mL, 0.2 mL, 0.4 mL, 0.8 mL, 1.2 mL, 1.6 mL, and 2.0 mL) and distilled water (2.0 mL, 1.8 mL, 1.6 mL, 1.2 mL, 0.8 mL, 0.4 mL, and 0 mL) were added sequentially to 7 test tubes, and 2 mL of 58 mM L^−1^ p-aminobenzenesulfonic acid and 7 mM L^−1^ α-naphthylamine were added to test tube No.4. After shaking well, it was kept at 25 °C for 20 min. Measures of 1 mL of p-aminobenzenesulfonic acid and 1 mL of α-naphthylamine were added, respectively, and kept at 25 °C for 20 min after shaking well. Subsequently, the absorbance was measured at 530 nm and the standard curve was drawn. A measure of 1 g of the sample was ground into homogenate with 65 mM L^−1^ phosphate buffer (pH = 7.8) and centrifuged at 8000 r for 10 min at 4 °C. A measure of 0.5 mL of the supernatant was added with 0.5 mL of phosphate buffer and 0.1 mL of 10 mM L^−1^ hydroxylamine hydrochloride solution and incubated at 25 °C for 20 min; then, 1 mL of 58 mM L^−1^ p-aminobenzenesulfonate and 1 mL of 7 mM L^−1^ hydroxylamine hydrochloride solution were added in order. Then, 1 mL of 58 mM L^−1^ p-aminobenzenesulfonic acid and 1 mL of 7 mM L^−1^ α-naphthylamine were added sequentially, shaken well, and then reacted at 30 °C for 30 min. An equal volume of trichloromethane was added to extract the pigment, and then centrifuged at 10,000 r for 3 min, and the absorbance was measured at 530 nm. The H_2_O_2_ content (μmol g^−1^ FW) in plant samples was quantified following the method described by Velikova et al. [43]. To each of the six tubes, 300 μM L^−1^ of H_2_O_2_ solution and 0.1% TCA (trichloroacetic acid) were added, followed by 0.5 mL of 10 mM PBS and 1 mL of 1 M KI (potassium iodide). After shaking well, the tubes were allowed to stand at 28 °C for 1 h. Subsequently, the absorbance values were measured at 390 nm and a standard curve was drawn. Approximately 0.5 g of the sample was homogenized in 5 mL of 0.1% TCA centrifuged and 0.5 mL of the supernatant was mixed with 0.5 mL of 10 mM pH = 7 potassium phosphate buffer and 1 mL of 1 M KI, and the absorbance was measured at 390 nm.

#### 2.3.4. Superoxide Dismutase (SOD), Peroxidase (POD), Catalase (CAT), and Ascorbate Peroxidase (APX)

Fresh leaf samples were used to determine the activity of several enzymes. The SOD activity was assessed following the nitrogen blue tetrazolium (NBT) method at 560 nm, as described by Giannopolitis and Ries [44]. Similarly, the POD activity was determined by the absorbance at 470 nm using the guaiacol solution, in accordance with the methodology detailed by Omran [45]. Additionally, the CAT activity was evaluated by monitoring changes in absorbance at 240 nm, as outlined by Aebi [46]. The APX activity was measured by monitoring the decrease in absorbance at 290 nm following the approach developed by Nakano and Asada [47]. For SOD, 0.5 g of fresh leaves were taken and ground and homogenized in 5 mL of 100 mM L^−1^ potassium phosphate buffer (pH 7.8) containing 0.1 mM L^−1^ EDTA, 0.1% Triton X-100, and 2% polyvinylpyrrolidone. Pyrrolidone was ground and homogenized in 5 mL of 100 mM L^−1^ potassium phosphate buffer (pH 7.8). The extracts were centrifuged at 15,000× *g* for 15 min at 4 °C and the supernatant was taken for the assay. The total volume of 3 mL of assay solution containing 50 mM L^−1^ sodium carbonate/sodium bicarbonate buffer (pH 9.8), 0.1 mM L^−1^ EDTA, 0.6 mM L^−1^ epinephrine, and the enzyme was measured at 560 nm and expressed as one unit of enzyme activity (u) for 50% inhibition of NBT photochemical reduction. For POD, 0.5 g of fresh leaves were ground in 3 mL of 0.1 M L^−1^ phosphate buffer (pH 7.0), and the extract was centrifuged at 18,000× *g* for 15 min at 4 °C. The supernatant was added with 0.5 mL of 0.1 M L^−1^ guaiacol solution and reacted for 15 min at 30 °C in a water bath; then, the absorbance per minute was recorded at 470 nm after shaking with 2.5 mL of distilled water. POD activity was expressed as (u g^−1^). CAT activity was determined by the change in absorbance at 240 nm by taking about 0.5 g of fresh leaves and adding 1 mL PBS buffer for grinding. The extract was centrifuged at 10,000× *g* for 10 min at 4 °C to remove the supernatant for measurement at 240 nm. The amount of enzyme decreased by 0.1 in 1 min at OD240 was expressed as one unit of enzyme activity (u). For APX, 0.5 g of fresh leaves were taken and ground by adding 5.0 mL of pre-cooled extraction buffer (0.1 mM L^−1^ EDTA, 1 mmol L^−1^ ascorbic acid, and 2% PVPP) at 4 °C, and the extract was centrifuged at 12,000× *g* for 30 min at 4 °C. A measure of 0.1 mL of the supernatant was taken and added to 2.6 mL of reaction buffer (0.1 mM L^−1^ EDTA and 0.5 mM L^−1^ ascorbic acid), and the enzymatic reaction was initiated by the addition of 0.3 mL of 2 mM L^−1^ H_2_O_2_, which was immediately mixed, and timing was started. The absorbance value of the reaction system at 290 nm was recorded from 15 s after the initiation and every 30 s for 2 min.

#### 2.3.5. Leaf Area Index (LAI), Photosynthetic Potential, Leaf Area Decreasing Rate (LAD), and Net Assimilation Rate (NAR)

At the heading and filling stage, five hills from each plot were selected to determine the dry matter weight. The leaf area was measured by length×width×0.75. The leaf area index (LAI) was calculated after the measurement.
Leaf area index (LAI) = total leaf area/land area,
photosynthetic potential = 1/2 (L1 + L2) (t2 − t1),
where L1 and L2 are leaf area (m^2^), and t1 and t2 are time (d) measured before and after treatment.
Leaf area decreasing rate (LAD) = (LAI2 − LAI1)/(t2 − t1),
where LAI1 and LAI2 are the leaf area index measured before and after treatment, and t1 and t2 are the time (d) measured before and after treatment, respectively.
Net assimilation rate (NAR) = [ln (LAI2) − ln (LAI1)]/(LAI2 − LAI1) × (W2 − W1)/(t2 − t1),
where LAI1 and LAI2 are the leaf area indices, respectively. W1 and W2 are dry matter masses (g m^−2^), and t1 and t2 are times (d) measured before and after treatment.

#### 2.3.6. Sucrose Synthase (SuSase), ADPG Pyrophosphorylase (AGPase), Starch Synthase (StSase), and Starch Branching Enzyme (SBE)

SuSase, StSase, AGPase, and SBE activities were determined according to the activity assay kit (Solarbio Science & Technology, Beijing, China). For SuSase, 0.1 g of seeds were ground and extracted in 1 mL of extraction solution. The extract was centrifuged at 10,000 g, 4 °C for 10 min to obtain a clear supernatant. The supernatant was then mixed with reagents according to the manufacturer’s instructions, incubated in a water bath at 80 °C for 20 min, and allowed to cool before undergoing further centrifugation at 12,000 rpm at room temperature for an additional 10 min. The supernatant was taken, and the OD value of the solution at 480 nm was recorded. One unit of SuSase activity was defined as the production of 1 μg of sucrose per minute from 1 g of sample in the reaction system. For StSase, the protocol followed was essentially similar, with the addition of 200 μL of the supernatant to the solution according to the attraction kit instructions, followed by the measurement of the OD value at 340 nm. One unit of StSase activity was defined as 1 nM NADPH produced per minute of 1 g of sample. The assay for AGPase involved grinding 0.1 g of seeds in 1 mL of extraction solution, centrifuging the extract at 10,000 g at 4 °C for 10 min, and adding 20 μL of the supernatant to the working solution. The OD value was measured at 340 nm, and one unit of AGPase activity was defined as the production of 1 nM NADPH per minute for 1 g of sample. For SBE, 0.1 g of seeds were ground and extracted in 1 mL of extract solution. The extract was centrifuged at 15,000 g, 4 °C for 15 min, followed by the subsequent addition of reagents. The OD values were measured at 660 nm, and one unit of SBE activity was defined as a 1% reduction in iodine blue value per minute for 1 g of sample weight.

#### 2.3.7. Determination of Grain-Filling Parameters

Approximately 100 main stem panicles were marked on each plot for the day’s draw. Each plot-marked panicle was randomly selected at a time, starting 7 days after flowering and spaced at 7 d intervals until maturity. The remaining seeds on the primary branching peduncle of the upper three branches of each panicle, except for the second seed at the top, were collected as superior grains, and the remaining seeds on the secondary branching peduncle of the lower three branches, except for the first seed at the top, were collected as inferior grains. A portion of the seed grains was frozen in liquid nitrogen for 1 min and then stored at −80 °C for the determination of the activities of key enzymes in the sucrose–starch pathway, and the rest was dried at 80 °C for about 48 h to constant weight. The grain-filling rate (the change of grain weight with days of tasseling) was fitted by the Richards equation [48], which was expressed as follows: W = A(1 + Be^−kt^)^−1/N^,
where A is the maximum grain growth; B is the initial value parameter; K is the grain growth rate parameter; and N is the curve formation parameter. The mean filling rate (G_mean_), time to maximum filling rate (T_max_G), and maximum grain weight (W_max_) were calculated from the fitted equations.

#### 2.3.8. Rice Yield and Yield Component

At rice maturity, six representative plants were randomly sampled from each experimental plot to evaluate various yield components. These included the effective number of panicles, number of grains per panicle, seed-setting rate, and thousand grain weight. Harvested rice plants from a standardized area of 10 m^2^ were subjected to threshing to determine the final grain yield.

### 2.4. Statistical Analysis

Data were analyzed according to the experimental design using SPSS 22.0 software (IBM Corp., Armonk, NY, USA). Use descriptive statistics to examine the average and standard error of measured variables. One-way ANOVA was employed to analyze the effect of biochar on the measured parameters. Two-way ANOVA was used to identify the interaction effect of biochar treatment and nitrogen application on the parameters of measurement. The least significant difference (LSD) test was used at *p* < 0.05. Correlation analysis and graphing were performed using Origin2021 (OriginLab Inc., Northampton, MA, USA).

## 3. Results

### 3.1. Concentration of Na^+^, K^+^, and Na^+^/K^+^ Ratio

The effects of different biochar treatments on Na^+^ concentration, K^+^ concentration, and Na^+^/K^+^ of rice leaves in soda saline–alkaline soils are shown in Figure 1. Biochar significantly decreased rice leaf Na^+^ concentration and Na^+^/K^+^, but significantly increased K^+^ concentration. Compared with N0B0, Na^+^ concentration decreased by 30.16%, 40.21%, 50.27%, and 61.90% in N0B1, N0B2, N0B3, and N0B4 treatments, respectively; meanwhile, Na^+^ concentration decreased by 19.18% in N225B1, N225B2, N225B3, and N225B4 treatments, respectively, compared with N225B0, 34.25%, 42.92%, and 45.21%, respectively (Figure 1A). Regardless of N application, biochar treatment significantly increased the K^+^ concentration in rice leaves, as shown in B4 > B3 > B2 > B1 > B0, and the differences between the biochar treatments and B0 reached the significant level (*p* < 0.05), but the differences between B3 and B4 were not significant (Figure 1B). Na^+^/K^+^ showed a tendency of decreasing with the biochar treatments under both N0 and N225 (Figure 1C), and the differences between the biochar treatments were evident. The differences between the treatments and the B0 treatment all reached significant levels (*p* < 0.05); among them, B1, B2, B3, and B4 were reduced by 49.89–86.15% compared with the B0 treatment. The two-way ANOVA showed that there was a significant interaction effect between nitrogen fertilizer (N) and biochar (B) treatments on Na^+^/K^+^.

### 3.2. Soluble Sugar, Soluble Protein, and Proline Concentration

As shown in Figure 2, the biochar treatments showed the effect of B0 > B1 > B2 > B3 > B4 on soluble sugars. Under N0, the soluble sugar content of B1, B2, B3, and B4 treatments at the heading stage was reduced by 19.40%, 37.93%, 38.36%, and 41.38%, respectively, compared with the B0 treatment; meanwhile, at the filling stage, the soluble sugar content of B1, B2, B3, and B4 treatments at the heading stage was reduced by 1.49%, 4.48%, 19.40%, and 23.88%; biochar under N225 treatment was 77.67% and 32.23% lower than N0 treatment at heading and filling stages (Figure 2A). The soluble protein content of rice leaves at heading stage under biochar treatment was higher than that of B0 treatment under N0, and the differences between the B2, B3, and B4 treatments and the B0 treatment reached a significant level (Figure 2B); at the filling stage, only the differences between B2 and B0 treatments reached a significant level (*p* < 0.05) under N0 and N225. However, the high-biochar treatment (B4) significantly reduced the soluble protein content of rice leaves under N225 at the filling stage. Biochar treatment significantly increased the proline content of rice leaves at heading stage and the filling stage, and the proline content under both N0 and N225 treatments showed B2 > B3 > B4 > B1 > B0 at heading stage; the proline content of the B1, B2, B3, and B4 treatments at filling stage increased by 6.82–31.02% under the N0 treatment compared with the B0 treatment, and increased by 9.48–35.40% (Figure 2C). The two-way ANOVA showed significant interaction effects of nitrogen fertilizer (N) and biochar (B) treatments with soluble sugar and soluble protein contents in soda saline–alkaline soils.

### 3.3. MDA, (O_2_^−^) and H_2_O_2_ Concentration

The effects of biochar on MDA, O_2_^−^, and H_2_O_2_ of rice leaves in soda saline–alkaline land is shown in Figure 3. Under both N0 and N225 treatments, rice MDA at heading stage and filling stage showed B0 > B1 > B2 > B4 > B3; under N0, biochar treatment reduced rice MDA by an average of 28.00% and 4.95% at heading stage and filling stage; compared with N225B0, at the heading stage, N225B1, N225B2, N225B3, and N225B4 decreased by 28.07%, 31.05%, 32.64%, and 31.46%, respectively; they decreased by 30.79%, 32.35%, 42.91%, and 40.97%, respectively, at the heading stage compared to N225B0 (Figure 3A). Biochar treatments were not consistent in their effects on rice leaf O_2_^−^ content, which generally showed that rice leaf O_2_^−^ content under B1, B2, and B3 treatments showed an increasing trend in both reproductive stages; the B2 treatment especially differed from the B0 treatment to a significant level in the heading stage (Figure 3B); however, the high-biochar treatment (B4) significantly decreased rice O_2_^−^ content (*p* < 0.05) under the N225 treatment at the heading stage and the N0 and N225 treatments at the filling stage, with O_2_^−^ content under the treatments (*p* < 0.05). Compared with N0B0, the H_2_O_2_ content of rice leaves in N0B1, N0B2, N0B3, and N0B4 treatments was reduced by 48.25−78.60% and 53.13−70.84% at heading stage and filling stage, respectively, and by 6.63−63.78% and 11.59−78.45%, respectively, under N225 (Figure 3C). The results of two-way ANOVA showed significant interaction effects of nitrogen fertilizer (N) and biochar (B) treatments with on MDA, O_2_^−^, and H_2_O_2_ contents of soda saline–alkaline paddy lands.

### 3.4. SOD, POD, CAT, and APX Concentrations

As shown in Figure 4, biochar treatment increased the activity of leaf SOD at both fertility periods of rice in soda saline–alkaline land (Figure 4A), where the B2 treatment differed from the B0 treatment to a significant level (*p* < 0.05) at both fertility periods of rice. Biochar treatment reduced the leaf POD activity of rice at both fertility stages with B0 > B1 > B2 > B3 > B4; at heading stage and filling stage under N0, biochar treatment reduced the leaf POD activity by an average of 20.65% and 2.95%, with the differences between the B2, B3, and B4 treatments and the B0 treatment reaching a significant level (*p* < 0.05); under N225 treatment, the leaf POD activity was reduced by 8.30% and 5.99% (Figure 4B), with B3 and B4 treatments differing from B0 at a significant level (*p* < 0.05). Biochar treatment increased rice leaf CAT activity (Figure 4C) at heading stage; the differences between the biochar treatments and B0 were not significant at N0, and only the B2 treatment at the filling stage differed significantly (*p* < 0.05) from B0 at N225. The differences between the B2 and B3 treatments and B0 were significant (*p* < 0.05) at both reproductive stages. The interaction effect of biochar and nitrogen fertilizer on CAT was significant. Compared with N0B0, N0B1, N0B2, N0B3, and N0B4 treatments reduced leaf APX activity at heading stage of rice by 3.1%, 5.24%, 5.24%, and 10.14%, respectively, but the differences among treatments were not significant; while at the filling stage, it was reduced by 21.88%, 48.02%, 56.53%, and 60.79%, respectively, and the differences between N0B2, N0B3, and N0B4 differed significantly (*p* < 0.05) from N0B0. Under N225, B1, B2, B3, and B4 reduced the APX activity of rice leaves at heading stage by 19.96%, 22.42%, 24.87%, and 24.80%, respectively, and at the filling stage by 15.54%, 31.29%, 46.83%, and 49.89%, respectively, as compared with that at B0 (Figure 4D); and the differences between B2, B3, and B4 and B0 all reached significant levels (*p* < 0.05).

### 3.5. LAI, Photosynthetic Potential, LAD, and NAR

The effects of biochar treatment on the photosynthesis-related parameters of rice are shown in Table 3. Biochar increased the leaf area index (LAI) of rice in soda saline–alkaline paddy field at both heading stage and filling stage. Under N0, the LAI of rice at both heading stage and filling stage was B4 > B3 > B2 > B1 > B0, and the differences between B3 and B4 and B0 reached the significant level; under N225, the LAI of rice at heading stage reached the significant level of B2, B3, and B4, and the differences of all treatments of biochar and B0 reached the significant level (*p* < 0.05) at the filling stage; B0 treatments reached a significant level here (*p* < 0.05). Biochar significantly increased the photosynthetic potential of rice from the heading stage to the filling stage by 9.38%, 40.63%, 143.75%, and 218.75% in N0 for B1, B2, B3, and B4, respectively, compared with B0; while in N225, B1, B2, B3, and B4 increased by 10.37%, 22.41%, 24.07%, and 16.18%, respectively, compared with the B0 treatment. Biochar reduced the LAD of rice from the heading stage to the filling stage, and the difference in leaf area decrement rate of biochar only in B3 and B4 treatments reached a significant level (*p* < 0.05) from B0 treatment in N0 and N225. Biochar increased NAR in rice from heading stage to filling stage, but the effect of increase was not significant in B1 treatment, and the difference reached a significant level (*p* < 0.05) under B2, B3, and B4 treatments; the biochar treatments increased by an average of 131.82% and 51.85% in the N0 and N225 conditions. The two-way ANOVA indicated that there was a statistically significant (*p* < 0.05) between nitrogen fertilizer (N) and biochar (B) treatments on LAI and photosynthetic potential of rice leaves.

### 3.6. Activity of Key Enzymes of the Sucrose–Starch Metabolic Pathway

The effects of biochar on the activities of key enzymes of the sucrose–starch metabolic pathway are shown in Figure 5. The activities of biochar on grains SuSase (Figure 5A), AGPase (Figure 5B), StSase (Figure 5C), and SBE (Figure 5D) showed single-peak curves. Biochar significantly increased SuSase activity in rice grains. Under superior grains, N0B1, N0B2, N0B3, and N0B4 treatments increased by 2.30%, 11.49%, 14.94%, and 6.90%, respectively, compared to B0, while in inferior grains it increased by 3.23%, 5.38%, 6.88%, and 11.83%, respectively; under N225, B1, B2, B3, and B4 increased compared to B0 in superior grains by 5.14–17.52%, while inferior grains increased by 7.55–24.48% (Figure 5A). Biochar significantly increased AGPase activity in rice grains, and grains reached maximum activity at B3 treatment. Biochar treatment increased the AGPase activity of grains by an average of 45.00% and 30.56% under N0 and N225 conditions (Figure 5B). The activity of StSase in rice grains reached its maximum at 14 days after tasseling. Biochar treatment increased StSase by 10.57–24.29% and 11.76–18.26% in superior and inferior grains, respectively, under N0, and by 5.19–17.50% and 7.12–15.74%, respectively, under N225 (Figure 5C). The activities of SBE in the grains after biochar application all showed B3 > B4 > B2 > B1 > B0. Under N0, the biochar treatments increased the activities of SBE in rice superior grains by 9.60%, 26.10%, 39.46%, and 28.60%, respectively, while the inferior grains increased by 21.51–43.21%, and under N225, the activities of B1, B2, B3, and B4 in the superior grains increased by 10.36%, 20.54%, 31.25%, and 28.13% compared to B0, while inferior grains increased by 8.40%, 18.28%, 28.45%, and 19.40%, respectively (Figure 5D).

### 3.7. Grain-Filling Parameters of Superior and Inferior Grains

The grain weights and filling rates of superior and inferior grains are shown in Figure 6. The inferior grains reached the maximum filling rate later than the superior grains and the grain weight was lighter. Nitrogen fertilizer treatment reduced grain weight and grain-filling rate in the early stage of filling; in the middle and late stages of filling, grain weight and filling rate were higher in the nitrogen fertilizer treatment (N225) than in the nitrogen-free treatment (N0). Biochar increased the average grain-filling rate (G_mean_) and maximum grain weight (W_max_) and prolonged the time for grains to reach the maximum filling rate (T_max_G) (Table 4). Under N0, the G_mean_, T_max_G, and W_max_ of the grains showed B4 > B3 > B2 > B1 > B0, and the G_mean_, T_max_G, and W_max_ of the superior grains of B1, B2, B3, and B4 were increased by 8.06–10.16%, 6.63–14.18%, and 9.62–14.36%, respectively, compared to B0; and the inferior grains were increased by 3.43–11.42%, 5.41–11.36%, and 6.08–18.60%, respectively. The differences between the biochar treatments and the B0 treatment reached significant levels (*p* < 0.05). Under N225, the superior grains G_mean_, T_max_G, and W_max_ increased by 3.67–6.42%, 7.48–19.24%, and 6.22–13.05% in B1, B2, B3, and B4, respectively, compared with B0; meanwhile, the inferior grains increased by 0.7–4.50%, 3.35–13.93%, and 2.02–14.21%, respectively. Two-way ANOVA showed that there was a significant interaction effect (*p* < 0.05) between nitrogen fertilizer (N) and biochar (B) treatments on rice grains G_mean_, T_max_G, and W_max_.

### 3.8. Yield and Yield Components

As shown in Table 5, biochar significantly increased the rice yield in a soda saline–alkaline paddy field. Under N0, the number of rice panicles increased by 11.76%, 20.58%, 32.35%, and 34.83%, while the number of grains in panicles increased by 6.69%, 14.38%, 25.20%, and 25.97%, respectively, compared to B0. Under N225, the number of rice panicles increased by 34.09–59.80%, and the number of grains in panicles increased by 10.63–23.29%, respectively; in the various treatments of biochar, compared to B0, this increased by 34.09–59.80%, while the number of panicles increased by 10.63–23.29%. The differences between the biochar treatments and the B0 treatment were all significant (*p* < 0.05). Biochar increased the seed setting rate of rice, and only B3 and B4 differed significantly (*p* < 0.05) from the B0 treatment at N0; the differences among the other treatments were not significant. The addition of biochar increased the thousand-grain weight of rice, but the differences between the biochar treatments and the B0 treatment were not significant. Of the rice yield, 13.33%, 33.33%, 55.67%, and 65.67% were increased in the B1, B2, B3, and B4 treatments, respectively, under N0; and 10.56%, 15.37%, 22.78%, and 19.81% of rice yield were increased under the N225, respectively. The differences between the biochar treatments and the B0 treatment reached the significant level (*p* < 0.05). Two-way ANOVA showed that there was a significant interaction effect (*p* < 0.05) between nitrogen fertilizer (N) and biochar (B) treatments on the number of panicles, the number of grains in panicles, and the yield of rice in soda saline–alkaline soils.

### 3.9. Correlation and Principal Component Analysis of Na^+^/K^+^, Physiological Indicators, Enzymes Activities, Filling Parameters, and Yield

As shown in Figure 7, rice yield was significantly and positively correlated (*p* < 0.05) with soluble protein, proline, SOD, POD, CAT, SuSase, AGPase, StSase, SBE, LAI, NAR, G_mean_, T_max_G, and W_max_, but significantly and negatively correlated (*p* < −0.05) with soluble sugar; MDA and O_2_^−^ were significantly negatively correlated (*p* < −0.05). The positive correlations (*p* < 0.01) between SuSase, AGPase, StSase, and SBE and the filling parameters were highly significant. In the principal component analysis (PCA), PC1 and PC2 of osmoregulatory substances, antioxidant enzymes, photosynthetic capacity, enzymes activities, as well as filling parameters and yield in rice organs contributed 79.3% and 11.4% to the total variance, respectively. The results showed that biochar with nitrogen fertilizer had effects on physiological indicators, key enzymes of starch synthesis and filling rate of rice in soda saline–alkaline soils.

## 4. Discussion

### 4.1. Relationship between Biochar and Rice Ion Balance in Soda Saline–Alkaline Paddy Soils

High sodium (Na^+^) concentration not only hinders potassium (K^+^) uptake by the root system, but also disrupts physiological and biochemical processes in crop cells; these multifaceted effects not only impede the uptake of essential nutrients by the crop, but also result in substantial yield reductions [4,6]. Decreasing Na^+^ levels and increasing K^+^ content in plant tissues are crucial strategies for mitigating saline–alkaline stress and reclaiming saline–alkaline land [28]. Numerous researchers have demonstrated that biochar can reduce Na^+^ influx into crop cells and mitigate salt-stress-induced crop losses, owing to its unique physicochemical properties and strong adsorption capacity [11,29,49]. Biochar possesses a multitude of functional groups, particularly carboxyl groups, on its surface, which enhance the soil’s ability to adsorb metal ions [50]. In our research, the addition of biochar in soda saline–alkaline paddy soil with or without N fertilizer application significantly reduced the Na^+^ concentration (Figure 1A) and Na^+^/K^+^ ratio (Figure 1C) of rice leaves in soda saline–alkaline paddy field, and the degree of decrease in Na^+^ concentration and Na^+^/K^+^ ratio was more pronounced as the amount of biochar application increased; meanwhile, the leaf K^+^ concentration was significantly increased by the application of biochar (Figure 1B). In this study, we also found that the improvement of ionic balance of soda saline–alkaline paddy field organs by biochar application in combination with nitrogen fertilizer was significantly better than that of biochar application alone (Figure 1). Song et al. indicated that the increase in K^+^ concentration and K^+^/Na^+^ ratio in the organ is an important factor to promote the growth and yield enhancement of peanut in saline–alkaline soil [8]. The findings suggest that biochar has an important role in maintaining the ionic balance of organs and abating Na^+^ toxicity in soda saline–alkaline paddy fields. The reasons may be as follows: (i) the higher adsorption capacity of biochar allows for the rapid removal of Na^+^ from soil solution [51,52]; (ii) biochar is considered to be a direct source of minerals, including Ca^2+^ and Mg^2+^ (Table 2), which are responsible for the displacement of Na^+^ from the exchange sites [53,54,55]; (iii) Na^+^ fixation and increased K^+^ uptake resulted in ionic homeostasis in cellular tissues (Figure 1) and attenuation of oxidative stress (Figure 3); and (iv) application of biochar optimized the root system of rice in saline–alkaline land and enhanced the uptake capacity of nutrients and water [23]. Furthermore, the study also observed a notable increase in K^+^ accumulation and a decrease in Na^+^/K^+^ ratio, with significantly higher effects seen in the biochar paired with nitrogen fertilizer treatment compared to biochar alone (Figure 1). This difference could be attributed to the inherently low fertility and poor structure of saline–alkaline paddy soil, where additional nitrogen fertilizer application enhances N accumulation and utilization capacity in rice organs [56]. The combination of biochar and N fertilizer led to a significant rise in K^+^ concentration in rice organs in soda saline–alkaline land, highlighting a key factor contributing to the enhanced efficacy of biochar paired with nitrogen fertilizer over biochar alone.

### 4.2. Relationship between Biochar and Rice Physiological Indicators in Soda Saline–Alkaline Paddy Soils

Excessive salt concentration in saline–alkaline soil leads to osmotic stress as the crop’s ability to absorb water is reduced or even extravasated [57,58]. Ion toxicity destabilizes the structure of plant cell membranes, leading to massive accumulation of ROS and thus oxidative damage [59,60]. Hafez et al. [61] conducted a study where biochar was applied to dry field soil, resulting in improved stability of barley cell membrane structure and antioxidant system. This was achieved through the reduction in soluble sugars and proline content, as well as a decrease in the activity of CAT and POX enzymes. The research also indicated a significant enhancement in barley’s drought tolerance. Mehmood et al. demonstrated that the addition of modified biochar led to a notable decrease in soluble sugars, soluble proteins, and proline contents in NaCl-stressed soybean plants [62,63]. Additionally, Wang et al. [64] observed that biochar helped alleviate the impact of salt stress on cotton plants by enhancing both the antioxidant enzyme system and non-enzymatic system in cotton leaves during a pot experiment. In line with previous findings, the present study found that the addition of biochar to soda saline–alkaline paddy soil significantly reduced the soluble sugar content, and the decreasing trend was more pronounced with the increase in the amount of biochar applied (Figure 2A); whereas, biochar significantly increased the content of soluble protein (Figure 2B) and proline (Figure 2C), which was most significantly increased by the treatment of 1.5% (*w*/*w*) (B2) (Figure 2C); meanwhile, biochar significantly increased soluble protein (Figure 2B) and proline content (Figure 2C), with the most significant increasing trend at 1.5% (*w*/*w*) treatment (B2); and biochar significantly decreased MDA (Figure 3A), H_2_O_2_ (Figure 3C), POD (Figure 4B), and APX content (Figure 4D) at the heading stage and the filling stage of rice in soda saline–alkaline land. Meanwhile, rice leaf O_2_^−^ (Figure 3B) under low charcoal application treatments (0.5%, 1.5%, and 3.0% *w*/*w*) showed an increasing trend in both reproductive stages, while high-biochar treatment (4.5% *w*/*w*) significantly reduced rice O_2_^−^ content under the N225 treatment at heading stage and the N0 and N225 treatments at filling stage. The reason for this may be that rice under soda saline–alkaline stress was subjected to a level of stress that far exceeded its own regulation, and the application of biochar alleviated the saline–alkaline stress and restored the plant’s regulatory capacity. Chaganti et al. [65] reported that saline–alkaline soil EC was significantly reduced by high biochar application (5.0% *w*/*w*), which abrogated saline–alkaline stress. Salar et al. [66] also reported that 10% and 20% (*w*/*w*) of biochar significantly reduced O_2_^−^ content in soybean leaves under heavy NaCl stress; this may be the reason for the significant reduction in O_2_^−^ content of rice in soda saline–alkaline land under high-biochar treatment in the present study. We also found that the SOD (Figure 4A) and CAT (Figure 4C) contents of rice in soda saline–alkaline land plants increased significantly after biochar application, with the most significant effect at 1.5% (*w*/*w*) charcoal application. Our previous study also showed that the addition of peanut hull biochar significantly reduced the ABA and MDA contents and increased the stability of cell membrane structure in soda saline–alkaline land rice leaves [19]. The results showed that biochar was able to reduce osmotic stress and oxidative injury in soda saline–alkaline land and enhance the stability of cell membrane structure. This may be due to the following reasons: firstly, the unique physicochemical properties of biochar and the large number of oxygen-containing functional groups on its surface improved the soil physicochemical properties of saline–alkaline soil, increased soil fertility, and enhanced the nutrient and water uptake capacity of the root system of rice in saline–alkaline land [23,32,67]; second, the addition of biochar significantly reduced Na^+^ concentration and Na^+^/K^+^ of rice leaves in soda saline–alkaline land, which abrogated the ionic toxicity of rice in soda saline–alkaline paddy field (Figure 1). Also, in this study, it was found that the ability of biochar in combination with nitrogen fertilizer to abate saline–alkaline barriers in soda saline–alkaline stress rice field was significantly better than the application of biochar alone. The reason for this is that the application of N fertilizer enhanced the K^+^ concentration in saline–alkaline stress rice organs (Figure 1B), reduced saline–alkaline stress (Figure 2, Figure 3 and Figure 4), and enhanced the stability of the cell membrane structure of rice under saline–alkaline stress [56,68]; meanwhile, our preliminary 6-year long-term experiments demonstrated that the pairing of biochar and N fertilizer was more conducive to the establishment of reasonable soil C/N, improving the soil structure and microbial community characteristics, and significantly increasing soil nutrient effectiveness [17,32].

### 4.3. Relationship between Biochar and Rice LAI, Photosynthetic Potential and NAR in Soda Saline–Alkaline Paddy Soils

Ion toxicity in saline–alkaline stress causes premature leaf senescence, reduces leaf area, decreases net assimilation rate, and limits photosynthetic capacity [69,70,71]. Feng et al. [72] found that the application of biochar to saline–alkaline stress soil was able to increase the biomass of soybean by increasing leaf photosynthetic area. The addition of biochar promoted the formation of photosynthetic pigments and photosynthetic products and increased the accumulation of carbohydrates in cotton leaves in acidic soils [18]. The study of Hafez et al. [61] also reported that the increase in the number of leaves, chlorophyll concentration, and the relative water content of leaves by biochar was an important factor in mitigating the effects of drought stress on barley. In this study, we found that biochar significantly increased LAI, photosynthetic potential, and NAR in soda saline–alkaline land rice at both fertility stages and from heading to filling stage (Table 3). The results of this study showed that biochar has an important role in enhancing the production capacity of photosynthetic material in rice in soda saline–alkaline land. The reasons are as follows: (i) biochar increased crop nutrient uptake by improving saline–alkaline soil structure and nutrient effectiveness, and promoted crop photosynthetic activity and chlorophyll synthesis [19,73]; (ii) biochar application significantly decreased Na^+^ concentration (Figure 1A) of rice leaves in soda saline–alkaline land and Na^+^/K^+^ (Figure 1C) were significantly reduced and K^+^ concentration was significantly increased (Figure 1B); (iii) biochar reduced soluble sugars (Figure 2A), MDA (Figure 3A) and H_2_O_2_ content (Figure 3C) and enhanced the antioxidant capacity (Figure 4) of rice in soda saline–alkaline land, as well as maintaining the hormone balance of the organ [73]. Furthermore, Zhu et al. [74] discovered that increasing the application of nitrogen fertilizer could enhance the activity of oxidative enzyme complex (OEC), photosynthetic performance, and the ability to scavenge reactive oxygen species, effectively mitigating the negative impacts of drought stress on yucca. Zhang et al. [75] also noted that N fertilizer could stimulate root growth and biomass accumulation in spring maize, leading to greater net photosynthetic rate, transpiration rate, and stomatal conductance, ultimately resulting in increased yield. These findings highlight the significant role of nitrogen fertilizer in improving LAI, photosynthetic potential, and NAR in rice under soda saline–alkaline stress when combined with biochar, compared to biochar treatment alone. Correlation and principal component analysis also showed that osmoregulatory substances and antioxidant enzyme activities were significantly and positively correlated with LAI and NAR (Figure 7).

### 4.4. Relationship of Biochar with Key Enzyme Activities of Starch Synthesis and Grain-Filling Parameters in Soda Saline–Alkaline Paddy Soils

Saline–alkaline stress limits the source bank capacity of rice through ion toxicity and other factors, changes the chain length structure of starch, reduces the transcript levels of genes for key enzymes of the sucrose–starch metabolism pathway, and inhibits the activity of key enzymes of starch synthesis [27,76]. Gong et al. reported under a potting assay that the moderate amount of biochar increased the expression of encoded genes and increased the activities of StSase, SBE, and DBE in rice grains [29]. In this study, it was observed that biochar significantly enhanced the activities of SuSase, AGPase, StSase, and SBE in superior and inferior grains during the filling stage of rice in soda saline–alkaline land (Figure 5). This is in accordance with the results of previous research. The results suggest that biochar has an essential role in increasing the activities of key enzymes for starch synthesis in soda saline–alkaline paddy field. Biochar increased the activities of key enzymes for starch synthesis of rice in soda saline–alkaline paddy soil grains for several reasons. Firstly, biochar absorbs organic molecules from the soil, promoting the polymerization of these molecules and catalyzing their activity on its surface, leading to the formation of organic matter and an increase in soil fertility [77]. Secondly, biochar significantly reduced Na^+^ concentration (Figure 1A) and Na^+^/K^+^ ratio (Figure 1C), while also increasing antioxidant enzyme activities (Figure 4) and mitigating the osmotic stress (Figure 2) of rice in soda saline–alkaline land. Lastly, biochar improved the photosynthetic potential and net assimilation rate of rice in soda saline–alkaline paddy field (Table 3), enhancing its overall photosynthetic capacity. Additionally, our study revealed that biochar had a greater impact on enhancing the activity of inferior grains of rice in soda saline–alkaline soil conditions compared to superior grains. This could be attributed to the higher ABA content and stronger osmotic stress observed in the superior grains during filling, as highlighted in previous research [78,79], and the inferior grains showed higher sensitivity to enzymatic activities of α-amylase, β-amylase, and SPS, as reported by Zang et al. [80].

Numerous studies have demonstrated that saline–alkaline stress, including ionic toxicity and osmotic stress, disrupts chloroplast structure, reduces chlorophyll content, inhibits photosynthesis transport and utilization [7,18,20], and hinders panicle differentiation, ultimately reducing grain filling efficiency [21,22,23]. Wang et al. discovered that biochar enhances dry matter accumulation and maize filling rate in saline soils by improving microbial environment and soil physicochemical properties [30]. In this experiment, we found that biochar significantly enhanced G_mean_, T_max_G, and W_max_ in superior and inferior grains of rice in soda saline–alkaline land, and the increase in inferior grains was better to that in superior grains (Figure 6, Table 4). Our results suggest that biochar has the potential to increase the grain-filling parameters of rice in soda saline–alkaline paddy fields. Biochar improves the grain-filling characteristics of rice in soda saline–alkaline land for the following reasons: (i) the inclusion of biochar activates the oxidative action of volatiles and surface functional moieties, which enhances the soil’s ability to absorb nutrients and optimizes the structure and uptake function of the rice root system [23,32,67]; (ii) biochar application significantly reduced Na^+^ concentration (Figure 1A) and Na^+^/K^+^ (Figure 1C) and increased the K^+^ concentration (Figure 1B) of rice leaves in soda saline–alkaline land, abating ion toxicity; (iii) biochar modulated the content of osmoregulatory substances in soda saline–alkaline land (Figure 2) and improved the stability of cell membrane structure (Figure 3) and antioxidant enzyme activities (Figure 4); (iv) biochar increased the photosynthetic potential and NAR of rice leaves in soda saline–alkaline land and enhanced their photosynthetic material production capacity (Table 3); (v) the addition of biochar increased the activity of key enzymes of the sucrose–starch metabolism pathway of rice in soda saline–alkaline land grains (Figure 5). Meanwhile, our study also found that the grain-filling parameters of the treatment with biochar and N fertilizer were better than that of biochar alone (Figure 6, Table 4). The reason was that N fertilizer could increase the accumulation and transportation of nutrients from stem to panicle through NSC, which significantly increased the grain-filling rate and grain weight [81]. In addition, the results of correlation and principal component analysis also indicated significant positive correlations between SuSase, AGPase, StSase, and SBE activities and grain-filling parameters (Figure 7).

### 4.5. Relationship between Biochar and Soda Saline–Alkaline Paddy Soils Yield and Its Components

Saline–alkaline stress negatively affects the source–reservoir relationship in rice by enhancing osmotic stress and ionic toxicity, which inhibits grain filling and leads to lower yields. There are many studies confirming that biochar can alleviate the negative effects of saline–alkaline stress in crop growth [12,30,58]. Previous studies have reported that the addition of corn stover biochar (40 t ha^−1^) improved soil physicochemical properties and significantly increased corn yield in saline–alkaline soil [30]. Similarly, Zhao et al. [82] added biochar to soda saline–alkaline land and found that biochar increased the male ear weight and the 100-kernel weight of maize, which in turn increased yield. Biochar dosages of 6 and 12 t ha^−1^ increased wheat yields by 49.6–54.7% and maize yields by 49.2–56.7% in saline trials [83]. In the experiment, we found that biochar significantly increased the yield of rice in a soda saline–alkaline paddy field (Table 5), with the increase in the number of panicles and number of grains in panicles being the key factors for the increase in the yield of rice in soda saline–alkaline land. The mechanism of biochar to improve rice yield in soda saline–alkaline land can be summarized as follows: (i) biochar, due to its high specific surface area and strong capacity of adsorption, released nutrients into the soil, reduced Na^+^ concentration of rice in soda saline–alkaline land organs (Figure 1A), increased K^+^ concentration (Figure 1C), and mitigated the effects of ionic toxicity on soda saline–alkaline [63]; (ii) the addition of biochar reduced MDA (Figure 2A) and H_2_O_2_ content (Figure 2C) and increased antioxidant enzyme activities of rice leaves in soda saline–alkaline land, alleviating osmotic stress and oxidative stress in soda saline–alkaline paddy soil (Figure 4), while protecting the cellular integrity of rice tissues (Figure 3); (iii) biochar significantly increased the LAI, photosynthetic potential, and NAR of rice in soda saline–alkaline fields and enhanced its photosynthetic capacity (Table 3); (iv) the activities of starch synthase enzymes in soda saline–alkaline paddy fields were significantly enhanced by biochar application (Figure 5), which promoted starch formation and increased the rate of grain filling (Table 4). Moreover, the results of correlation analysis also showed that oxidative stressors, sucrose–starch metabolizing enzyme activities and filling parameters were significantly and positively correlated with yield (Figure 7).

Our previous studies showed that biochar application significantly reduced BD, ESP, Ece, and SARe, increased soil CEC, TP, Ks, nutrient effectiveness, and relative abundance of bacterial communities, and optimized root structure and uptake function of rice in a soda saline–alkaline field; this is also one of the important factors for improving the physiological indicators, photosynthetic material production, and irrigation characteristics of soda saline–alkaline paddy soil and hence increasing the yield of rice in soda saline–alkaline land after the application of biochar in combination with nitrogen fertilizer [17,19,23,32,52].

## 5. Conclusions

A two-year field experiment demonstrated that incorporating biochar into soda saline–alkaline rice fields effectively alleviated ion toxicity, osmotic stress, and oxidative damage in rice plants. The application of biochar led to an increase in the leaf area index of rice, enhanced leaf photosynthetic rate, and improved the production of photosynthetic substances. Furthermore, biochar with nitrogen fertilizer significantly boosted the activities of key enzymes in the sucrose–starch metabolic pathway of rice in soda saline–alkaline paddy fields, resulting in accelerated grain filling and ultimately higher rice yield. These findings offer novel insights into the use of biochar to mitigate soda saline–alkaline stress, enhance rice yield in such challenging environments, and support the sustainable and healthy development of rice in saline–alkaline paddy fields cultivation.

## Figures and Tables

**Figure 1 plants-13-02237-f001:**
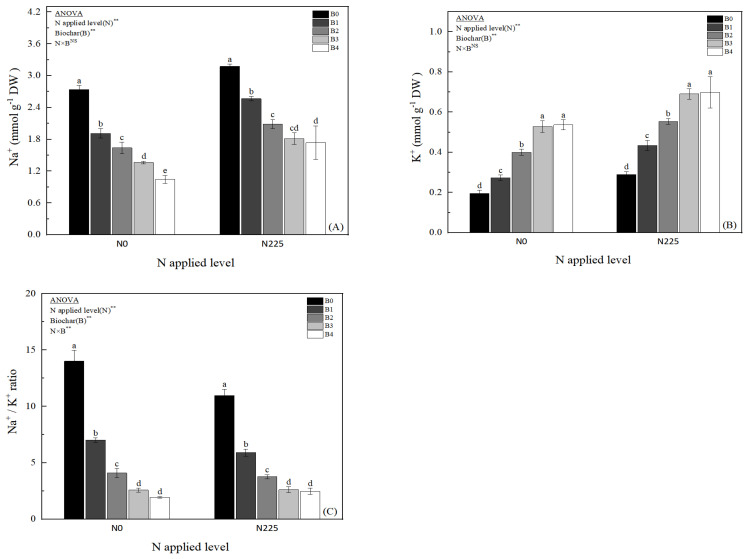
Effect of biochar on Na^+^ concentration (**A**), K^+^ concentration (**B**), and Na^+^/K^+^ ratio (**C**), B0, B1, B2, B3, and B4 represent 0%, 0.5%, 1.5%, 3.0%, and 4.5% biochar levels, respectively. N0 and N225 represent no N application and conventional N application levels, respectively. The data in the figure are averaged over the two study years. Note: ** denotes significant at the 0.01 level, and NS denotes not significant. Lowercase letters (a–e) on the upper side of the error line indicate differences between biochar treatments (*p* < 0.05).

**Figure 2 plants-13-02237-f002:**
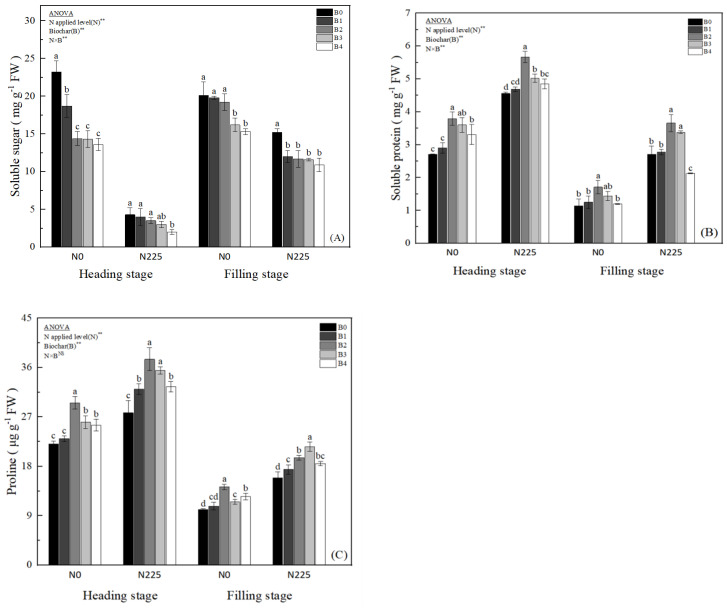
Effect of biochar on soluble sugar (**A**), soluble protein (**B**), and proline (**C**). B0, B1, B2, B3, and B4 represent 0%, 0.5%, 1.5%, 3.0%, and 4.5% biochar levels, respectively. N0 and N225 represent no N application and conventional N application levels, respectively. The data in the figure are averaged over the two study years. Note: ** denotes significant at the 0.01 level, and NS denotes not significant. Lowercase letters (a–d) on the upper side of the error line indicate differences between biochar treatments (*p* < 0.05).

**Figure 3 plants-13-02237-f003:**
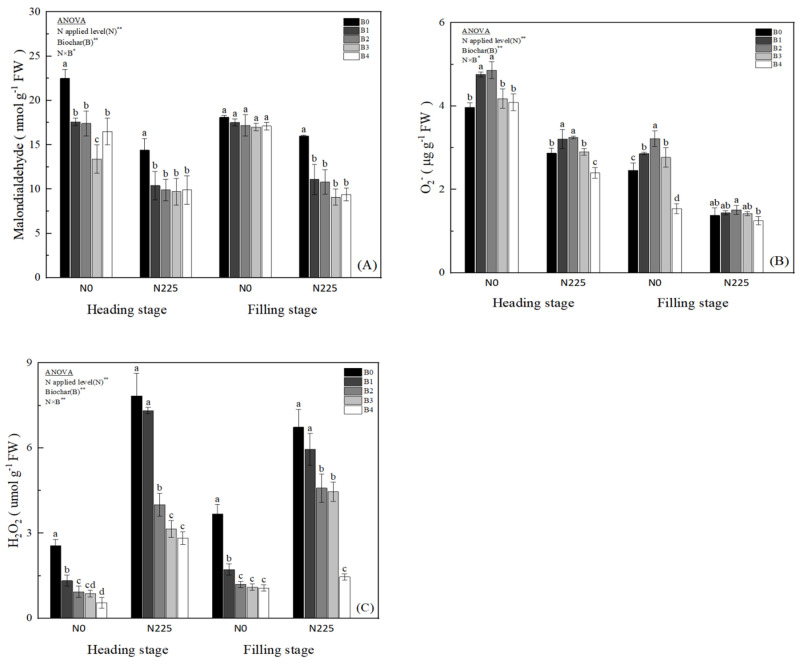
Effect of biochar on MDA (**A**), O_2_^−^ (**B**), and H_2_O_2_ (**C**). B0, B1, B2, B3, and B4 represent 0%, 0.5%, 1.5%, 3.0%, and 4.5% biochar levels, respectively. N0 and N225 represent no N application and conventional N application levels, respectively. The data in the figure are averaged over the two study years. Note: * and ** denote significant at the 0.05 and 0.01 levels, respectively. Lowercase letters (a–d) on the upper side of the error line indicate differences between biochar treatments (*p* < 0.05).

**Figure 4 plants-13-02237-f004:**
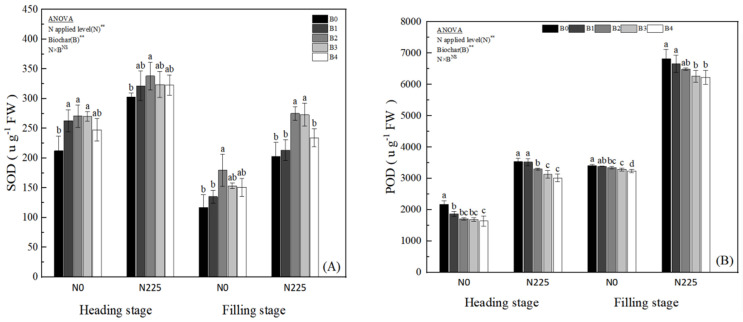
Effect of biochar on SOD, (**A**) POD (**B**), CAT (**C**), and APX (**D**). B0, B1, B2, B3, and B4 represent 0%, 0.5%, 1.5%, 3.0%, and 4.5% biochar levels, respectively. N0 and N225 represent no N application and conventional N application levels, respectively. The data in the figure are averaged over the two study years. Note: * and ** denote significant at the 0.05 and 0.01 levels, respectively, and NS denotes not significant. Lowercase letters (a–d) on the upper side of the error line indicate differences between biochar treatments (*p* < 0.05).

**Figure 5 plants-13-02237-f005:**
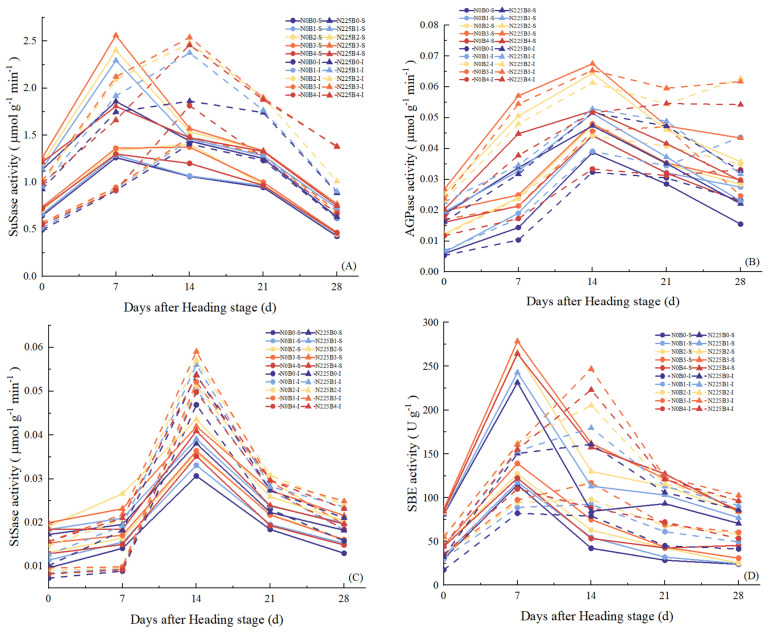
Effect of biochar on SuSase (**A**), StSase (**B**), AGPase (**C**), and SBE (**D**). B0, B1, B2, B3, and B4 represent 0%, 0.5%, 1.5%, 3.0%, and 4.5% biochar levels, respectively. N0 and N225 represent no N application and conventional N application levels, respectively. SuSase−sucrose synthase; AGPase−adenosine diphosphate glucose pyrophosphorylase phosphorylase; StSase−starch synthase; SBE−starch branching enzyme.

**Figure 6 plants-13-02237-f006:**
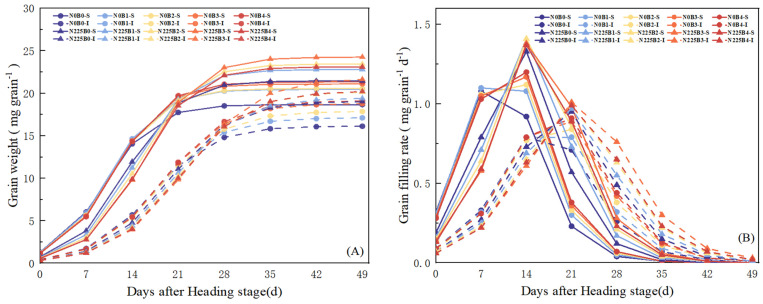
(**A**,**B**) The grain weight and filling rate of superior and inferior grains under different biochar treatments. B0, B1, B2, B3, and B4 represent 0%, 0.5%, 1.5%, 3.0%, and 4.5% biochar levels, respectively. N0 and N225 represent no N application and conventional N application levels, respectively.

**Figure 7 plants-13-02237-f007:**
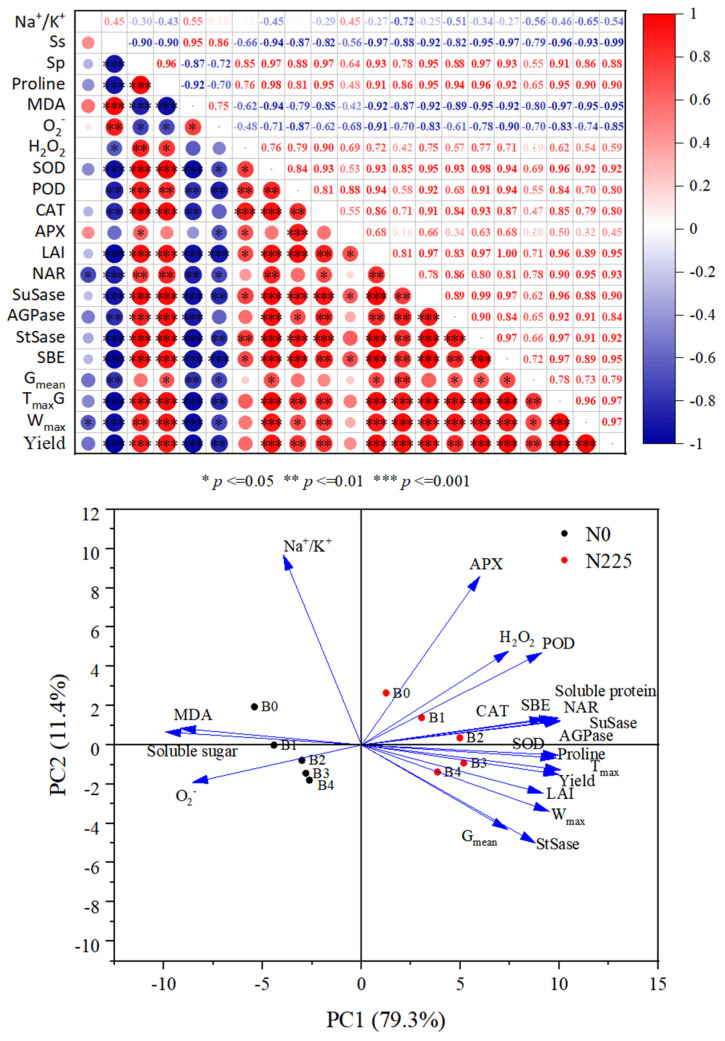
Correlation and principal component analysis of Na^+^/K^+^, physiological indicators, activity of key enzymes of starch synthesis, filling parameters, and yield. Ss−soluble sugar; Sp−soluble protein; MDA−malondialdehyde; SuSase−sucrose synthase; AGPase−adenosine diphosphate glucose pyrophosphorylase phosphorylase; StSase−starch synthase; SBE−starch branching enzyme; LAI−leaf area index; NAR−net assimilation rate; G_mean_−mean grain-filling rate; T_max_G−the time taken to reach the maximum grain-filling rate; W_max_−grain weight of reaching the maximum grain-filling rate.

**Table 1 plants-13-02237-t001:** Physicochemical properties of the soil [34].

Soil Properties (0–20 cm Soil Layers)	Value
Sand content (%)	23.13 ± 1.11
Silt content (%)	38.14 ± 1.31
Clay content (%)	37.60 ± 2.09
Bulk density (g cm^−3^)	1.61 ± 0.13
ECe (μs m^−1^)	24.08 ± 0.71
pH	10.10 ± 0.24
SARe (mmolc L^−1^)^1/2^	368.11 ± 4.03
ESP (%)	55.11 ± 2.17
CEC (cmol kg^−1^)	10.99 ± 0.34
Organic matter (%)	0.64 ± 0.04
Total N (g kg^−1^)	0.27 ± 1.11
Alkali-hydrolysable N (mg kg^−1^)	16.30 ± 1.11
Available P (mg kg^−1^)	9.13 ± 0.68
Available K (mg kg^−1^)	107.25 ± 5.68

Ece—electrical conductivity of soil saturation extract; SARe—sodium adsorption ratio of soil saturation extract; ESP—exchangeable sodium percentage; N—nitrogen; P—phosphorus; K—potassium.

**Table 2 plants-13-02237-t002:** Basic properties of biochar [34].

pH and Elemental Component	Biochar
pH	7.94 ± 0.32
CEC (cmol kg^−1^)	78.69 ± 11.32
EC (dS m^−1^)	7.88 ± 0.59
SA (m^2^ g^−1^)	7.41 ± 0.12
C (mg g^−1^)	540.64 ± 26.58
N (mg g^−1^)	15.93 ± 1.01
S (mg g^−1^)	6.85 ± 0.34
P (mg g^−1^)	0.74 ± 0.03
Mg (mg g^−1^)	0.25 ± 0.00
K (mg g^−1^)	12.53 ± 0.51
Ca (mg g^−1^)	2.01 ± 0.02
Na (mg g^−1^)	1.17 ± 0.21

CEC—cation exchange capacity; EC—electrical conductivity; C—carbon; N—nitrogen; S—sulfur; Mg—magnesium; P—phosphorus; K—potassium; Ca—calcium; Na—sodium.

**Table 3 plants-13-02237-t003:** Effects of different biochar treatments on leaf area index (LAI), photosynthetic potential, leaf area decreasing rate (LAD), and net assimilation rate (NAR).

N Applied Level	Biochar Applied Rates	LAI		Heading Stage to Filling Stage		
		Heading Stage	Filling Stage	Photosynthetic Potential (m^−2^ d)	Leaf Area Decreasing (LAI d^−1^)	NAR (g m^−2^ d^−1^)
N0	B0	0.35 ± 0.01 b	0.29 ± 0.01 d	0.32 ± 0.00 e	0.0135 ± 0.0003 a	0.11 ± 0.05 b
	B1	0.40 ± 0.01 b	0.30 ± 0.02 d	0.35 ± 0.01 d	0.0134 ± 0.0002 a	0.15 ± 0.03 b
	B2	0.51 ± 0.02 b	0.40 ± 0.02 c	0.45 ± 0.02 c	0.0113 ± 0.0013 b	0.25 ± 0.03 a
	B3	0.86 ± 0.03 a	0.70 ± 0.04 b	0.78 ± 0.03 b	0.0115 ± 0.0013 b	0.27 ± 0.03 a
	B4	1.04 ± 0.23 a	0.99 ± 0.02 a	1.02 ± 0.12 a	0.0117 ± 0.0001 b	0.35 ± 0.12 a
N225	B0	2.65 ± 0.21 b	2.16 ± 0.12 c	2.41 ± 0.14 d	0.0279 ± 0.0063 a	0.27 ± 0.09 b
	B1	2.80 ± 0.11 ab	2.51 ± 0.16 b	2.66 ± 0.04 c	0.0242 ± 0.0034 a	0.30 ± 0.08 b
	B2	3.02 ± 0.17 a	2.87 ± 0.17 a	2.95 ± 0.08 ab	0.0217 ± 0.0022 ab	0.43 ± 0.03 a
	B3	3.08 ± 0.10 a	2.91 ± 0.17 a	2.99 ± 0.12 a	0.0166 ± 0.0040 b	0.47 ± 0.04 a
	B4	2.97 ± 0.07 a	2.69 ± 0.07 ab	2.80 ± 0.07 b	0.0167 ± 0.0039 b	0.44 ± 0.07 a
ANOVA N applied level (N) Biochar (B) N × B						
		**	**	**	**	**
		**	**	**	**	**
		*	**	**	NS	NS

The data in the table are averaged over the two study years. Note: * and ** indicate significant at the 0.05 and 0.01 levels, respectively, NS indicates not significant, and the same letter in the same column indicates no significant difference at the 0.05 level.

**Table 4 plants-13-02237-t004:** Filling parameters of superior and inferior grains.

N Applied Level	Biochar Applied Rates	G_mean_ (mg (100 Grain d)^−1^)		T_max_ G (d)		W_max_ (mg (100 Grain)^−1^)	
		Superior	Inferior	Superior	Inferior	Superior	Inferior
N0	B0	82.05 ± 0.22 d	57.96 ± 0.18 e	9.80 ± 0.06 e	16.64 ± 0.18 c	933.58 ± 2.83 c	807.46 ± 4.73 d
	B1	88.69 ± 0.59 bc	59.95 ± 0.17 d	10.45 ± 0.10 d	17.54 ± 0.38 b	1023.42 ± 11.46 b	856.58 ± 14.17 c
	B2	88.53 ± 1.03 c	61.02 ± 0.68 c	10.74 ± 0.04 c	18.20 ± 0.24 a	1028.07 ± 10.99 b	894.40 ± 2.92 b
	B3	90.14 ± 0.94 ab	63.60 ± 0.60 b	10.99 ± 0.17 b	18.45 ± 0.16 a	1054.96 ± 11.26 a	941.46 ± 11.76 a
	B4	90.39 ± 1.02 a	64.58 ± 0.53 a	11.19 ± 0.03 a	18.53 ± 0.12 a	1067.62 ± 12.08 a	957.64 ± 14.41 a
N225	B0	89.81 ± 2.46 b	64.84 ± 0.30 b	13.10 ± 0.17 d	19.38 ± 0.04 e	1073.31 ± 23.23 d	953.51 ± 4.36 d
	B1	93.99 ± 0.57 a	65.28 ± 0.15 b	14.08 ± 0.19 c	20.03 ± 0.15 d	1140.08 ± 9.54 c	972.75 ± 8.54 c
	B2	94.99 ± 0.74 a	65.83 ± 1.30 b	14.83 ± 0.22 b	20.82 ± 0.21 c	1172.80 ± 15.11 b	1011.00 ± 15.68 b
	B3	95.58 ± 1.14 a	67.76 ± 1.07 a	15.62 ± 0.15 a	22.08 ± 0.21 a	1213.33 ± 17.47 a	1088.96 ± 5.74 a
	B4	93.12 ± 1.38 a	65.65 ± 0.13 b	15.22 ± 0.33 ab	21.12 ± 0.14 b	1155.97 ± 15.34 bc	1014.59 ± 7.44 b
ANOVA N applied level (N) Biochar (B) N × B							
		**	**	**	**	**	**
		*	**	**	**	**	**
		**	**	**	**	**	**

G_mean_—mean grain-filling rate; T_max_ G—the time to reach the maximum grain-filling rate; W_max_—grain weight of reaching the maximum grain-filling rate. B0, B1, B2, B3, and B4 represent 0%, 0.5%, 1.5%, 3.0%, and 4.5% biochar levels, respectively. N0 and N225 represent no N application and conventional N application levels. Note: * and ** indicate significant at the 0.05 and 0.01 levels, and the same letter in the same column indicates no significant difference at the 0.05 level.

**Table 5 plants-13-02237-t005:** Effect of biochar on yield and components of rice in soda saline–alkaline paddy fields.

N Applied Level	Biochar Applied Rates	Panicle Number (10^4^ hm^−2^)	Spikelets per Panicle	Seed Setting Rate (%)	1000-Grain Weight (g)	Yield (t hm^−2^)
N0	B0	145.07 ± 7.83 c	59.80 ± 1.52 d	87.58 ± 2.38 b	27.14 ± 1.05 a	3.00 ± 0.20 d
	B1	162.13 ± 9.90 b	63.80 ± 1.21 c	89.83 ± 3.30 ab	27.95 ± 0.63 a	3.40 ± 0.10 c
	B2	174.93 ± 12.42 ab	68.40 ± 5.52 bc	91.02 ± 2.30 ab	28.84 ± 2.94 a	4.00 ± 0.03 b
	B3	192.00 ± 12.90 ab	74.87 ± 6.33 ab	91.93 ± 1.37 a	29.31 ± 2.59 a	4.67 ± 0.21 a
	B4	195.60 ± 11.03 a	75.33 ± 4.17 a	92.00 ± 1.21 a	29.72 ± 1.86 a	4.97 ± 0.15 a
N225	B0	187.73 ± 10.90 c	85.03 ± 3.45 c	92.50 ± 1.39 a	25.43 ± 0.99 a	5.40 ± 0.40 d
	B1	251.73 ± 10.45 b	94.07 ± 2.18 b	92.77 ± 4.08 a	25.77 ± 0.14 a	5.97 ± 0.06 c
	B2	296.53 ± 16.13 a	99.57 ± 5.34 ab	93.63 ± 2.68 a	26.04 ± 0.70 a	6.23 ± 0.16 b
	B3	300.80 ± 23.13 a	104.83 ± 10.29 a	94.88 ± 1.06 a	26.13 ± 0.72 a	6.63 ± 0.03 a
	B4	256.00 ± 21.35 b	102.03 ± 9.58 ab	93.41 ± 2.61 a	25.50 ± 1.28 a	6.47 ± 0.05 b
ANOVA N applied level (N) Biochar (B) N × B						
		**	**	**	**	**
		**	**	*	**	**
		**	**	NS	NS	*

B0, B1, B2, B3, and B4 represent 0%, 0.5%, 1.5%, 3.0%, and 4.5% biochar levels, respectively. N0 and N225 represent no N application and conventional N application levels. Note: * and ** indicate significant at the 0.05 and 0.01 levels, respectively, NS indicates not significant, and the same letter in the same column indicates no significant difference at the 0.05 level.

## Data Availability

The data that support the findings of this study are available from the corresponding author upon reasonable request.

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
