# Peer review of "Biochar Improves Yield by Reducing Saline–Alkaline Stress, Enhancing Filling Rate of Rice in Soda Saline–Alkaline Paddy Fields"

_plants, 2024, doi:10.3390/plants13162237_

Round 1
Reviewer 1 Report
Comments and Suggestions for Authors
Dear authors, i read with interest the manuscript, please see my comments on the PDF attached. I think the manuscript is good, very good results and discussion but the main criticism are related to the introduction and M&M. Following my suggestion i think the paper can be published in Plants journal.
Title to long, please reduce and make more short for readers

minor editing
Author Response
Thank you so much for your careful review and valuable advice for our paper. We have made detailed modifications in accordance with the suggestions and requirements proposed by you and the other reviewers. Please see the attachment.

Reviewer 2 Report
Comments and Suggestions for Authors
Dear Authors,
In my opinion, the manuscript is good, fits well with our previous studies. I have some comments, see the attached file below. Two main suggestions I have are: related to sodium, potassium concentration, and LSD test. These are necessary improvements to publish this manuscript.

Author Response

(The authors gave the same response as above.)

Round 2
Reviewer 1 Report
Comments and Suggestions for Authors
The authors performed a good job in the revision. In my opinion there is only one problem left to be improved, concerning the part of the introduction where in lines 83-87 they added a part concerning the where I would suggest adding an additional part concerning the possible mechanisms of action of biochar to counteract salt stress. Also I suggest adding more references in line 86-87 where the authors reported, "In addition, biochar holds promising potential as a soil amendment for the revitalization of salt-affected soils."
I suggest the authors to read the following paper:
https://doi.org/10.1016/j.heliyon.2024.e26526
https://doi.org/10.1111/jac.12132